# Extracellular DNA: Insight of a Signal Molecule in Crop Protection

**DOI:** 10.3390/biology10101022

**Published:** 2021-10-11

**Authors:** Ireri Alejandra Carbajal-Valenzuela, Gabriela Medina-Ramos, Laura Helena Caicedo-Lopez, Alejandra Jiménez-Hernández, Adrian Esteban Ortega-Torres, Luis Miguel Contreras-Medina, Irineo Torres-Pacheco, Ramón Gerardo Guevara-González

**Affiliations:** 1C. A. Biosystems Engineering, Campus Amazcala, Autonomous University of Queretaro, Carr. Chichimequillas-Amazcala Km 1 S/N, C.P., El Marques, Querétaro 76265, Mexico; jancarval@hotmail.com (I.A.C.-V.); inglauraclo23@gmail.com (L.H.C.-L.); ale.jhtsu@gmail.com (A.J.-H.); adrianesotorres@gmail.com (A.E.O.-T.); mcontreras.uaq@gmail.com (L.M.C.-M.); irineo.torres@uaq.mx (I.T.-P.); 2Molecular Plant Pathology Laboratory, Polytechnic University of Guanajuato, Cortazar 38496, Mexico

**Keywords:** eDNA, elicitors, hormesis, sustainable agriculture, DAMPs

## Abstract

**Simple Summary:**

Agriculture systems use multiple chemical treatments to prevent pests and diseases, and to fertilize plants and eliminate weeds around the crop. These practices are less accepted by the consumers each day, mostly because of the associated environmental, health, and ecological impact; thus, new sustainable green technologies are being developed to replace the use of chemical products. Among green technologies for agriculture practices, the use of plant elicitors represents an alternative with great potential, and extracellular DNA has shown beneficial effects on important production traits such as defence mechanisms, plant growth and development, and secondary metabolites production that results in yield increment and better-quality food. In this review, we reunite experimental evidence of the natural effect that extracellular DNA has on plants. We also aim to contribute a step closer to the agricultural application of extracellular DNA. Additionally, we suggest that extracellular DNA can have a biostimulant effect on plants, and can be applied as a highly sustainable treatment contributing to the circular economy of primary production.

**Abstract:**

Agricultural systems face several challenges in terms of meeting everyday-growing quantities and qualities of food requirements. However, the ecological and social trade-offs for increasing agricultural production are high, therefore, more sustainable agricultural practices are desired. Researchers are currently working on diverse sustainable techniques based mostly on natural mechanisms that plants have developed along with their evolution. Here, we discuss the potential agricultural application of extracellular DNA (eDNA), its multiple functioning mechanisms in plant metabolism, the importance of hormetic curves establishment, and as a challenge: the technical limitations of the industrial scale for this technology. We highlight the more viable natural mechanisms in which eDNA affects plant metabolism, acting as a damage/microbe-associated molecular pattern (DAMP, MAMP) or as a general plant biostimulant. Finally, we suggest a whole sustainable system, where DNA is extracted from organic sources by a simple methodology to fulfill the molecular characteristics needed to be applied in crop production systems, allowing the reduction in, or perhaps the total removal of, chemical pesticides, fertilizers, and insecticides application.

## 1. Introduction

Climate change constitutes a serious threat to the environment and all living organisms. In particular, numerous studies suggest serious consequences for the health of crop plants, affecting both the productivity and the quality of raw materials destined for the food industry [1]. Among the agriculturally important risks associated with a changing climate are drought, increased incidence of diseases and pests, decreased availability of water and pasture resources, forest fires, hailstorms, and, in general, biotic and abiotic stress that vegetal crops will have to overcome to maintain food production [2].

In this context, agricultural yields decline largely due to climatic changes, leading to the loss of optimal conditions for crop production, but also due to an increase in the incidence of pest-related losses [3,4,5,6,7,8]. By decreasing crop yields and decreasing global income derived from crop production [9], climate change thus increases food insecurity, as both agricultural and livestock production systems are threatened by changing patterns of rainfall and temperature, which causes malnutrition [10]. The use of chemical pesticides, fertilizers, and herbicides in agriculture has also generally decreased its acceptance due to its high environmental, health, and ecological costs, plus the loss of aggregated value of the non-organic crops [11,12].

In addition, current intensive agricultural practices, including land clearing, excessive and inefficient use of fertilizers and pesticides, irrigation, and the use of fossil fuels for agricultural machines, make agriculture a major contributor to greenhouse gas emissions [13], and maintaining this never-ending cycle contributes to environmental problems.

To manage the risk of climate change, knowledge of climate-smart agricultural practices (CSAPs) is imperative [14]. A transformative approach that reorients the agricultural sector to address climate change while ensuring sustainable food security, while food production is maintained or increased to meet the needs of a growing population and reducing negative environmental impacts from climate change and other factors, must be developed and applied [15].

Recently, research interest has focused on understanding the natural defence mechanisms of plants so new, natural agriculture treatments can be developed soon to cope with pests and diseases. In this area, a problem has been addressed concerning the selection of agronomically interesting traits in crops through common agricultural management, resulting in plant domestication. This artificial process has led to dramatic phenotypical changes in plants and unintended consequences for other traits may have ensued, such as metabolic trade-offs between increasing edible biomass and the activation of chemical defences as a reaction to biotic and abiotic stresses [16]. Theories concerning these kind of trade-offs assume the physiological cost of secondary metabolites and the necessary energy allocation [17]. Researchers have suggested that plant domestication may indirectly reduce plant defences due to increased allocation of plant resources to increased yield, supported by some experimental evidence [16,17,18,19,20,21].

In this context, the use of plant metabolism modifiers (PMFs) has been suggested as a potential treatment that suits most modern agricultural needs. PMFs have been labelled and classified in different ways, principally as biostimulants and elicitors and their use has shown to obtain higher fruit production, enhancement of plant growth, reduction in plant diseases occurrence, an increase in metabolites production as chlorophyll, carotenoids and, protein contents, as well as the production of some important enzyme activities as phenylalanine ammonia-lyase (PAL) and defence enzymes as catalase (CAT), superoxide dismutase (SOD), among others [22,23,24,25,26].

The results of several studies of elicitors/biostimulants application to plants have demonstrated that the dose of treatment shapes the effect in plant metabolism following a hormetic dose-response curve [27]. This behaviour suggests that the fine characterization of each elicitor/biostimulant molecule effect will help to determine a dose that elicits immune responses, while stimulating growth-developing signalling pathways to reduce the metabolic trade-off negative events [28].

One of the most promising emerging elicitors is extracellular DNA (eDNA), which has multiple roles in plant metabolism that will be later explained focusing on the possibility of being applied as a sustainable agricultural treatment. Here, we address the main concerns about the practical application of this potential treatment, technical challenges, and, finally, we propose a new use for this molecule as a general plant biostimulant, enhancing the possibility of a circular economy system.

## 2. Natural Conditions of eDNA Release and Sensing

Recently, a new condition of DNA has been identified as eDNA. This term refers to all DNA molecules located outside a cell [29,30]. The presence of eDNA in almost every environment results from multiple different processes of both active release from physiologically active cells and passive release from moribund or dead cells from all kinds of organisms [31,32]. eDNA has long been known as one of the most abundant molecules in almost every environment and, therefore, organisms have developed different biological roles for it, as a vector of horizontal gene transfer [33], as a source of nutrient and energy due to its content of nitrogen and carbon [29], and as a structural molecule for biofilms and extracellular neutrophil traps construction [34,35].

As eDNA can be diffused in the environment through various mechanisms, it can be found in diverse conditions that reflect multiple natural events [36]. For example, differences in eDNA fragments size can mean different cell death mechanisms. In the process of apoptosis, nucleases cleave and degrade DNA and produce low molecular weight species, including ladders [37]. In necrosis, the DNA is usually not cut following a pattern, and generally, the molecule exhibits higher sizes [38].

The concentration of eDNA molecules can reveal the distance or time passed since a biological event. Plants growing next to decomposing conspecifics would be exposed depending on the distance to lower or higher concentrations of eDNA, indicating the closeness to possible danger and the possibilities to survive [39]. Relevantly, the source, carriage, and deterioration of eDNA influence their physical state, regulating the mechanisms and effects of its interactions. Organisms have developed sensing and signal transduction mechanisms to respond to the presence of eDNA in specific situations, relevant for their survival [40].

In this review, we focus on vegetal organisms, as it has been seen that eDNA molecules play a major role in self and non-self-recognition and therefore in plant-host interactions. The next question researchers have addressed is how the organisms perceive eDNA and more importantly distinguish eDNA with specific molecular conditions.

To date, there is no knowledge about eDNA sensing and signal transduction mechanisms in plants, but in mammals, there is now a broad consensus that the receptors involved in nucleic acid-sensing have been identified as a specific family of Pattern Recognition Receptors (PRR) [41,42]. Being the first category of nucleic acid receptors, the Toll-like receptor family (TLR), specifically, TLR7, TLR8, and TLR9 recognize differences between nucleic acids molecules (Table 1) in animals [42,43]. These TLRs are localized within intracellular organelles and are involved in the recognition of microbial nucleic acids. The specificity of each TLR is determined mainly by one structural motif called Leucine-Rich Repeat (LRR), which is involved in the recognition of MAMPs [41].

Interestingly, it has been observed that DNA samples of different bacteria showed considerable differences in their potential to stimulate TLR-9 and this is correlated with the frequency of CG dinucleotides of the samples [42]. This is relevant in ecological interactions because differences in the content of CG DNA motifs have been addressed between prokaryotic and eukaryotic organisms and even between genera of bacteria. This is the first hint of an eDNA receptor with the capacity to identify differences in the origin of eDNA.

In the case of plants, some hypotheses have been discussed, Bhat and Ryu [44] attribute the perceptions of eDNA to four possible mechanistic scenarios: (a) the presence of some specific membrane-bound receptors of microbial DNA, consistent with what is known for mammals, (b) the presence of transporter channels on the membrane, leading eDNA through the cell membrane into the cytoplasm where they can trigger the proper signal cascades, (c) the internalization of eDNA molecules by vesicles into the cell, and (d) the presence of eDNA sensors that mediate the effect but localise intracellularly. Additionally, according to some authors [39,45,46], the eDNA has a species-specific inhibitory effect on the growth of plants, depending on the concentration, leading to the possibility of multiple recognition ways for multiple eDNA conditions.

In a recent study, the authors demonstrate that plant cells can sense eDNA distinguishing between self and non-self-DNA, revealed by the display of very different effects in plant transcriptome and different patterns of eDNA localization with non-self-DNA entering root tissues and cells and self-DNA remaining outside [47].

## 3. Self-eDNA as a DAMP

Several plant responses have been identified to have as a principal goal the maintenance of a high ecological diversity by inhibiting the development of single-species communities. These mechanisms are known as negative plant-soil feedback [48]. It has been observed that the development of juvenile specimens of a given species was limited in environments with a high density of adult organisms of the same species. This was explained as the possible enhancements by adult communities of highly specific pathogens living in the soil [49] or as the content of a molecule in the soil, where a plant grows, with an inhibitory effect on new plants of the same species. The last one refers more to the capacity of juveniles to distinguish highly similar organisms as “self” and respond to this stimulus.

The ability to distinguish “self” from “non-self” has been described as the most fundamental aspect of an immune system [50]. Recently, researchers have focused on identifying molecular mechanisms to explain self-inhibition. The suggested molecules for this role had to be highly conserved along with the organisms from all the kingdoms, but had to differ from one another in an identifiable way, depending on their origin, to allow for a species-specific inhibitory effect. One of the suggested molecules is DNA, as a highly species-specific, constantly present molecule. Supporting this hypothesis, several works have suggested the role of self eDNA as a danger signal molecule. The first report of it was made by Mazzoleni et al. [45] in a study where multiple species across different taxonomic groups, including plants, were exposed to respective eDNA. The experiments produced highly significant differences in responses to either self or non-self eDNA, treatments with self eDNA always resulted in a concentration-dependent growth reduction. Later, eDNA was applied in vitro and in vivo to different plant species at different concentrations. In these studies, the authors assessed germination, plant development, and plant growth. The root growth of all targeted species was significantly inhibited by fragmented self eDNA treatments in a concentration-dependent manner. These studies highlighted the importance of self eDNA fragmentation to produce biological reactions in plants. The fragmentation resembles the natural decomposition of biological tissues and the DNA degradation, resembling the decaying of a near specimen of the same species.

As with any other signal molecules, eDNA activates a hormetic dose-response in plants [27]. This means that different doses of the same molecule can cause toxicity or stimulation. In the case of self eDNA, the stimulation of several plant defence responses have been reported. Lima bean (*Phaseolus lunatus*) and maize (*Zea mays*) leaves responded to self eDNA with a plasma membrane potential depolarization and calcium signalling, both early response events preceding the build-up of chemical defence in plants [51]. The response had a linear behaviour with the concentration of treatment starting with <2 ug/mL for Lima bean and 12 ug/mL for maize up to 200 ug/mL of self eDNA. The self eDNA responses were compared to non-self eDNA application with plants showing no response to the latter [51].

Lettuce seedlings responses were also evaluated in the presence of self and non-self eDNA. All self eDNA treatments, except for the highest concentration (200 ug/mL), showed statistically significant changes in the hypomethylation levels of genomic DNA compared to deionized water in control [46]. Additionally, gene expression of early defence enzymes was evaluated in five-days-old lettuce seedlings, treated with self eDNA and non-self eDNA extracted from two plants with different phylogenetic distances to lettuce. The self eDNA treatment triggered a concentration-dependent effect in the expression of superoxide dismutase (*sod*), catalase (*cat*), and phenylalanine ammonia-lyase (*pal*) (2–200 ug/mL). Interestingly, non-self eDNA from *C. chinense* (phylogenetically close to lettuce) treatment activated the expression of *sod* and *cat* in the same levels as self eDNA (200 ug/mL) but *A. angustissima* (phylogenetically more distant from lettuce) did not affect the expression of these genes. The effect of eDNA application on *pal* gene expression was different from that of the other evaluated genes: *pal* expression was activated by *A. angustissima* eDNA (non-self eDNA) at a higher level than *C. chinense* (phylogenetically closer non-self eDNA) or even self eDNA. These results highlight the sensibility of plant immune responses related to the phylogenetic distance between species.

Durán-Flores and Heil [52] reported the effect of eDNA application on the formation of H_2_O_2_ and activation of MAPKs in the leaves of common bean. Self eDNA caused a significant (almost three-fold) increase in H_2_O_2_ compared to plants treated with non-self eDNA that had no detectable effect. Similarly, activation of MAPKs was detectable at 5 min and strongest at 30 min after self eDNA application. Non-self eDNA also showed MAPKs activation but at a lower level. Surprisingly, plants treated with self or non-self eDNA exhibited a decrease in infection rates after the inoculation of the phytopathogen *Pseudomonas syringae*.

Similar to these studies, Rassizadeh et al. [53], treated *Arabidopsis thaliana* five-weeks-old plants with self eDNA in different concentrations and evaluated its effect on signal transduction and metabolic pathways activation by specific transcripts measure, production of plant hormones, accumulation of H_2_O_2_, and resistance spectrum against pathogens with inoculation of the biotrophic bacteria *Pseudomonas syringae*, the oomycete *Hyaloperonospora arabidopsis*, the necrotrophic fungus *Botrytis cinerea* and the phloem sucking insect *Myzus persicae*. The results showed an up-regulation of transcription of genes involved in ROS signaling (OXIDATIVE SIGNAL-INDUCIBLE, *OXI1* and calcium signaling (CALMODULIN LIKE 37, *CML37*) but interestingly, the study showed no differential expression in some marker genes regulated by defense-related phytohormones, such as PATHOGENESIS-RELATED GENE 1 (PR-1) for salicylic acid, PLANT DEFENSIN 1.2 (PDF 1.2), VEGETATIVE STORAGE PROTEIN 2 (VSP2) and JASMONATE RESISTANT 1/JAR1) for jasmonic acid, ETHYLENE RESPONSE FACTOR 2 and 5 (ERF2 and ERF5) for ethylene.

As for the induction of resistance against pathogens, the self eDNA treatment induced plant resistance. The infection rates for pathogens were calculated by the measurement of infected leaves per plant and colony-forming units (CFU) for *P. syringae* infection, the number of conidiospores for *H. arabidopsis*, aphid population for *M. persicae* infestation, and necrotic lesion diameter caused by *B. cinerea* inoculation, all of them significantly reduced from control plants. All these results suggest that self eDNA treatment in *A. thaliana* plants induces defence signalling but not direct defence responses per se. Highlighting that although the signalling pathways of DAMP effects on plants are, at some level, studied, self eDNA may activate different metabolic pathways than that expected from a common DAMP recognition. Additionally, these authors, showed that self eDNA mediated defence activation leads to a broad range-resistance against diverse biotic stresses [53].

By these reports, the role of self eDNA as a damage-associated molecular pattern (DAMP) has been identified, similar to systemin [54], extracellular ATP [55], and *At*Pepl [56]. DAMPs have been described as aberrantly located endogenous molecules indicating self-damage and therefore danger leading to the activation of defence responses [57]. As immune system responses have been characterized, tested with multiple doses, vegetal species, and applying times, researchers started considering the self eDNA treatment as a possible agricultural application to elicit an early defence response from crops in a “vaccine-like” mechanism, as described by Quintana-Rodriguez et al. [58] and Ferrusquia-Jimenez et al. [59].

New studies have focused on understanding the effect of self eDNA in plants in a more global way. Barbero et al. [60] identified as an effect of self eDNA treatment in tomato plants a negative regulation of genes related to gene ontology terms such as metabolic and biosynthetic processes of Myo-inositol (>100-fold enrichment), nitric oxide metabolic and biosynthetic processes (>70-fold enrichment), biosynthetic processes of ROS, cell wall, jasmonic acid, and sucrose transport (>47 fold enrichment). Additionally, several genes were identified as upregulated and related to the following gene ontology terms: oxygen transport, defence against Gram-negative bacteria, and lactate biosynthetic processes (>66-fold enrichment), adenine biosynthetic, and metabolic processes, auxin influx, and cellular ion homeostasis processes (>33-fold enrichment).

Contrastingly, an up-regulation of *MPK3* and *OXI1* genes resulted from treatments with broccoli, citrus, bean, and maize eDNA. Similarly, CML37 levels of expression were up-regulated in *A. thaliana* plants treated with broccoli and bean eDNA only [60]. In this case, the non-self eDNA had neither pathogenic nor microbial origin to support the hypothesis of the eDNA effect as MAMP/PAMP but these results do support the phylogenetically closeness-related effect described earlier [46,52].

In a more recent study [47], DNA extracted from *Arabidopsis thaliana* plants and common herring (*Clupea harengus*) were applied to *A. thaliana* plants as self and non-self eDNA treatments. In self DNA responses, genes related to oxidative stress, toxic substances, and ions were up-regulated, involving genes encoding detoxification and anti-oxidation protective enzymes, while downregulating typical stress-responsive genes, as *PAD4* gene that has been related to pathogen resistance response mediated by TIR-NB-LRRs. This contrasts with the up-regulation of *PAD4* as an effect of non-self-DNA treatment, as well as several genes involved in systemic acquired resistance. These responses evidence differences in self and non-self-DNA activated mechanisms, consistent with DAMP-, P/MAMP- like responses, respectively.

Another important difference between responses is the up-regulation of genes related to ABA and jasmonic acid in the first hour after self eDNA treatment, while in non-self eDNA analysis an up-regulation of genes related to ABA and salicylic acid was revealed [47]. In self DNA response, an up-regulation of most of the genes belonging to the cytokinin oxidase/dehydrogenase family also suggests cytokinin-mediates processes affected, possibly involving cell cycle regulation, cell proliferation, and shoot and root development. This coupled with a down-regulation of gibberellins transport may be related to the growth inhibition as a result of defence mechanisms activation.

On one hand, the authors identified a remarkable differential gene expression in non-self eDNA treated plants compared to control plants, involving both biotic and abiotic stress-related genes accompanied by a hypersensitive response. On the other hand, a minor differential expression gene response to self-eDNA compared to control plants was identified relating to oxidative stress and the activation of the chloroplast gene expression, and the downregulation of stress-responsive genes [47]. The observed role of non-self eDNA in the activation of plant responses is described in the next section.

According to the observed reactions to different doses of self eDNA, Pontiggia et al. [61] suggested in their review that the plant must be able to discriminate between a physiological or a pathological event when sensing an accumulation of DAMPs, in this case, eDNA. Conditions as the time of formation, concentration, and distribution may help distinguish such events. In the case of a pathogen attack, plants need to respond quickly, intensively, and systemically to the danger. On the contrary, in a physiological event, plants may only need to activate local immune responses preventing a massive immune response with negative effects on plant growth. Even in pathological infections as those established by biotrophic or hemibiotrophic pathogens, early, local immune responses activated by specific doses of DAMPs, may be enough to attenuate pathological symptoms or prevent the switch between biotrophic to necrotrophic in pathogens, while maintaining the normal plant growth [62,63,64,65].

This potential agricultural application offers several advantages due to its species-selective mechanism and low concentration needed for an effect on the plant immune system, but mostly by its natural origin with low environmental, health, and ecological costs. Additionally, although there are no studies at a production scale of this natural mechanism, in theory, the raw material where DNA would be obtained from can be the same crop waste, lowering costs and becoming, this way, a more sustainable agricultural practice.

## 4. eDNA as a MAMP/PAMP

One of the most important steps for the plant immune system to work is the identification of the presence of a pathogenic organism to activate the proper responses to overcome the potential danger [66]. Different from the role of self eDNA, the perception of eDNA not only non-self but specifically coming from a prokaryotic source interpreted as the presence of a possible pathogenic microbe, has been suggested to activate defence mechanisms in plants [67,68,69]. Here we describe the role of microbial eDNA as a microbe-associated molecular pattern (MAMP) for the plant immune system.

The release of DNases from several microorganisms [70,71,72,73,74] and even herbivores [75] has been reported to induce pathogenesis-related genes and immune system activation in plants. Moreover, these DNA degrading enzymes appear to be highly involved in the pathogenic process. The role of DNases in pathogenesis has been little explored. Here, we link the presence of DNases in enzymes that pathogenic microbes secrete, with the role of microbe eDNA as a danger signal molecule for plants.

Wen et al. [76] reported the increase of infection by *N. haematococca* in pea roots treated with DNAse I in contrast with the root tips without DNAse I inoculated with the pathogen. This latter behaviour may be explained by the digestion of the eDNA present in root cap slime that has been identified to confer resistance in pea to microbial infections, but evidence suggests another explanation. The same experiments were carried out this time with excised root tips (without root cap slime) immersed directly within the treatment solutions. At 24 and 48 h after inoculation of root tips with fungal spores alone, hyphal growth was minimal. In root tips co-inoculated with the pathogen and DNAse I, a proliferation of hyphae on the surface and penetration into root tissue were evident within 24 h. In light of the latter result, we suggest that DNAse I could digest eDNA working as a signal molecule, allowing the pathogen to establish infection by avoiding the activation of the plant immune system.

As previously seen, the size of eDNA fragments is an important factor for the plant to recognize it as a danger signal. In this study, the authors also treated root tip with a slower degradation rate DNAse (BAL31) that generates genomic DNA fragments of 250 bp–6 kb after 24 h of treatment. By contrast, treatment with DNase I degrades DNA to fragments smaller than 250 bp within 2 h. Treatment with BAL31 resulted in a delay in infection establishment, compared with the response to DNase I, suggesting the presence of eDNA-mediated defence mechanisms activation.

There are multiple reports of defence mechanisms in plants triggered by pathogenicity-related-bacterial DNA. In 2009, Yakushiji et al. [67] treated *Arabidopsis thaliana* culture cells and leaves with different digestion enzymes-treated *E. coli* plasmids and measured the H_2_O_2_ production induced by treatments. The plasmid DNA trimmed in CG sequences caused a less intense activation of H_2_O_2_ production, suggesting the CG islands are important to this recognition mechanism. CpG DNA motifs are very rare in eukaryotic DNA but very common in prokaryotic ge1omes, and their role as MAMPs of the animal immune system is well known [77,78,79]. This DNA motif allows the organisms to discern between eDNA originated from eukaryotic cells and prokaryotic cells.

As a hypothesis, the total and rapid degradation of microbe eDNA by pathogen-released DNases causes a slower activation of the plant immune system, allowing the pathogen to establish an infection in a more discreet way for the immune system.

The treatment with a mixture of fragmented eDNA from different phytopathogenic organisms (*Phytophthora capsici*, *Fusarium oxysporum* and *Rhizoctonia solani*) also showed a protective effect in chili pepper against wilt and root rot disease [68]. In this study, both disease severity and plant mortality were measured with a reduction of 60% and 40% compared to the infected control, respectively. The plants immune system activation was also measured by total phenolics determination as well as by phenylalanine ammonium lyase and chalcone synthase gene expression analysis. The mixture of microbial fragmented eDNA treatment showed an immune system activation effect that can be related to a decrease in disease severity by the inoculated pathogenic complex in capsicum.

In another study, the application of short sequences of non-self eDNA with cytosine-phosphate-guanine oligodeoxynucleotide motifs in a concentration of 9.5 × 10^−5^ g/liter statistically reduced the lesions in leaves of wheat plants by the pathogenic fungus *Z. tritici*, showing a similar effect to a commercial fungicide [69].

With all these observations, the ability of plants to distinguish microbial eDNA from other kinds of eDNA is clear. The common hypothesis is that the content of CpG DNA motifs helps plant receptors to respond differently to microbial eDNA, and this can confer an ecological advantage to plants to identify near pathogens or beneficial bacteria. Thus, this natural mechanism can be seen as a potential agricultural application that could replace chemical pesticides at some level or completely. More tests are needed to complete the information about hormetic curves in multiple plant species. The concentration of applied eDNA needs to be carefully chosen because, as reviewed by van Butselaar and Van den Ackerveken [80], there are multiple molecular pathways in plants that cause a growth-immunity trade-off. It has been shown that the activation of immunity by salicylic acid (SA) signalling and jasmonic acid (JA) inhibit growth, involving auxin receptors [81], transport [82], and several transcription factors as TFBF1 [83]. Additionally, it has been shown that growth affects SA signalling. As the initial burst of growth slowly declines in the plant lifetime, SA signalling strengthens up [84].

Although the mentioned effects of eDNA application have been proved a great potential in agricultural management, each application must be carefully evaluated. A summary of the effect of applied eDNA experimentally evaluated is shown in Table 2.

## 5. Technical Challenges of eDNA Application as an Agricultural Treatment

The use of eDNA as an agriculture treatment could be questionable in terms of the cost–benefit of extraction, fragmentation, and application of eDNA, even knowing the ideal treatment concentration. In this section we propose a redefinition of the protocol needed for DNA extraction suitable for agricultural application by identifying the common steps in lab DNA extraction protocols and determining which steps are needed for this specific purpose, having in mind that the conditions needed for this goal are different from the conditions desired in lab extracted DNA. Usually, the lab DNA extraction aims not only for the significant quantity of nucleic acids, but for the high integrity and purity of the molecule, and some of the most expensive steps have these purposes. Otherwise, eDNA for agricultural treatments has shown a better immune response in plants in a fragmented state, and a high level of purity may be of lower importance [52].

### 5.1. DNA Extraction

As Duran-Flores and Heil in 2014 [39] have shown, the simple application of self leaves-homogenates in plants activates defence responses (similar to the so-called plant-derived biostimulants or PDBs). Although it has been suggested that eDNA inside homogenates is one of the main causes of the DAMP effect in treated plants, the homogenates contain several other molecules that not only lack the DAMP role but can be interfering in the eDNA sensing or lower eDNA stability. Additionally, the components of PDBs depend on several factors as plant genotype, organ, and stage of extraction, and environmental factors, that shape a metabolites expression, making the components highly variable [85]. In contrast, DNA would always be the same in a plant species. Here, we suggest that DNA extraction for agricultural purposes must be the simplest procedure that still activates secondary metabolism in plants and remains stable to allow an easy application. Possibly, the best DNA extraction method is a procedure between simple tissue homogenates and the traditional lab DNA extraction. We address the most important goals to achieve in DNA extraction as the elimination of most cellular debris in the sample that may contain lytic enzymes, biological contamination, and biological hazard for agricultural application of eDNA.

Another important factor to consider in the DNA extraction method is the generation of toxic or special waste disposal residues. In several DNA extraction protocols multiple chemicals are used and because of their toxicity they require specific waste disposition. For eDNA agriculture treatment to be a sustainable option, the waste of DNA extraction must be free of toxic or environmentally hazardous chemicals, as we discuss later.

To meet the agricultural purposes of DNA extraction, it is important to question every step applied in conventional lab DNA extraction that usually has different specific goals in terms of DNA integrity and purity. Conventional lab DNA extraction protocols require large or relatively large quantities of grams of tissue to be ground in a mortar with a pestle, and kits are generally either expensive or not readily available, particularly for researchers in developing and under-developed countries [86]. Therefore, extraction techniques have been diversified to fulfil all needs. Here, we describe the most common extraction methods divided into three main steps: (1) cell lysis, where different methods can be applied to achieve the rupture of the cell membrane and release of the cellular content; among several molecules, the DNA. (2) DNA extraction: consisting of the elimination of membrane debris and contaminant substances from the sample to reduce co-purification of unwanted components (proteins, lipids, polysaccharides, and polyphenols) and (3) DNA purification: in this last step, nucleic acids are isolated from remaining contaminants looking for a higher level of purity [87].

The first step of DNA extraction, cell lysis, has special importance due to the diversity of cells to lyse and the diversity of techniques available to achieve this. Initially, we suggest that the DNA for agricultural treatments may be extracted from decaying biological tissues. In DNA extraction protocols, younger tissues are advised to be used, this way better integrity is obtained in DNA results. In terms of the application of DNA in agricultural treatments, this would mean harvesting young leaves from crops or establishing new vegetal or microbiological cultures only for DNA extraction. This would increase the cost and complexity of the treatment. Instead, we suggest extracting DNA from pruning waste or other agricultural wastes, where tissue contains a level of damage in genetic material. In this case, extracted DNA would already have a level of damage or fragmentation, likely reducing the need for fragmentation procedure.

A wide variety of tissue lysis methods are available, usually, the selection is up to the budget, workflow, purification steps, target molecules of analysis, the quality of final extraction, and the tissue itself [86]. Islam et al. [88] highlight the importance of the tissue worked with. For agricultural purposes of DNA extraction, a general tissue lysis method must be designed, possibly by mixing cell lysing techniques. The most common cell lysis methods have been reviewed by Islam et al. [88] and Harrison [89], classified into the categories: mechanical, chemical (alkaline lysis and detergent lysis), physical (meaning all non-contact methods) and biological (lytic enzymes). The main advantages and disadvantages of each technique are described in Table 3.

In this review, we highlight the potential of physical methods coupled with mechanical and chemical techniques, mainly because of their adaptability to industrial scale and different tissues. Physical techniques are based on processes where voltage or acoustic waves are applied to several tissues aiming for the cell lysis. This is a highly versatile technique since voltage and sonication can cause cell rupture. We consider some factors that could affect efficiency depending on the kind of tissues treated. These factors may be adjusted based on specific biological DNA origin. Yield could depend on three factors:The amplitude of the voltage or intensity of soundThe diameter of the channel where the fluid with cells is passingThe application time of voltage or acoustic stimulus. The time can be managed by the pump, which varies the velocity of the fluid with cells.

The cell lysis step is commonly performed with the addition of extraction buffers used to maintain a stable environment and avoid molecule degradation by enzymes and pH changes. Often, the extraction buffers contain salt and organic compounds as Tris-HCl, EDTA, NaCl, SDS, as seen in KCl extraction buffer [94], DEB buffer [95], and Dellaporta buffer [96]. These organic compounds represent no environmental or health hazard according to their safety data sheets. Other commonly used components of extraction buffers are β-mercaptoethanol, a reducing agent that irreversibly denatures proteins, including degrading enzymes, by reducing disulphide bonds and destroying the native conformation required for enzyme functionality and Cetyltrimethylammonium bromide (CTAB), a cationic detergent of low cost that is generally the method of choice in DNA extractions without kits for a diverse kind of sample, but especially for plant tissues with high polysaccharides and other inhibitory substances [97,98,99]. Although very useful, these two chemicals have been reported as toxic by ecotoxicity tests [100,101,102] and are established as hazardous chemicals by their safety data sheet.

Some simple, safe handling methods with low equipment dependence and with the use of easily available non-toxic reagents have been developed for DNA extraction and proved to have the same performance as CTAB methods [92]. Briefly, these protocols consist of mixing the biological sample with an extraction buffer and applying a mechanical cell rupture technique, after this, a protein, and polysaccharides precipitation by the addition of salts such as sodium acetate and a centrifugation step, followed by a nucleic acid precipitation step with cold isopropanol [92]. Additionally, some experimental assays have successfully been performed to replace toxic chemicals such as β-mercaptoethanol, and this compound has been replaced by 1% sodium sulphite (NA_2_SO_3_) which is also cheaper [99].

Regarding the industrial scale of DNA extraction, we must calculate the scale of treatment production needed for each case: number of applications along the agricultural cycle, the desired effect on plants and thus the concentration of eDNA needed, also the plant species in which the treatment would be applied, how many plants are going to be treated and the yield of DNA extraction. Once the amount of eDNA needed is determined, we could think about the industrial escalation of the process. The two major processes needed to scale up in DNA extraction are tissue lysis and centrifugation, Piccino et al. mentioned in their review some industrial processes equivalent to lab-scale processes, such as rotor-stator type homogenizer or an industrial blender for tissue lysis and industrial centrifugation [103] and Islam et al. [88] have reviewed a list of commercially available mechanical cell lysis instruments. These are performed in already available industrial modules and their cost depends on the volume and power requested.

As an example, if a producer wants to elicit a tomato crop, and after the proper evaluations, the concentration to apply is 20 ppm of self eDNA, we estimate 10 mL of eDNA solution sprayed per plant per application. The producer would need about 200 L of eDNA solution per hectare (20,000 plants). At this concentration, this means we need to extract 4 g of DNA. Although this sounds simple, it needs a considerable amount of plant biomass. In the simple extraction protocol reported by Rodrigues et al., the yield is 15 ug of DNA per 25 mg of tissue and 1.5 mL of extraction buffer. If we use this as an example, the producer will need 6 Kg of tomato plant tissue and 400 L of extraction buffer to cover one whole application per hectare. Additionally, the costs can be lower if the eDNA is applied by normal irrigation systems.

One of the advantages of using plant tissue as the source of eDNA is that agriculture production has no problem obtaining it, for example, tomato (*Solanum lycopersicum*) wastes from greenhouse systems produce about 15 t ha^−1^ y^−1^ of fresh plant residues [104], of which 10.7 t ha^−1^ y^−1^ has been calculated to be from leaf biomass only [105]. Although the number of applications during the cultivation cycle is not established yet, the needed biomass can be easily covered using greenhouse plant waste. Here is an important logistic problem to address, because self eDNA would be obtained as the crops generate plant tissue waste through pruning, according to crop management practices [106], so initial applications should be made with eDNA obtained from previous crops or other sources, as reviewed in Section 4.

### 5.2. DNA Damaging/Fragmentation

As mentioned before, it has been shown that the integrity of the DNA as a molecule does not represent high importance for agricultural applications. Therefore, here we propose using decaying sources of DNA, which is likely to already have a considerable level of damage. This way, the process of fragmentation can be reduced or even eliminated from protocol. Either way, in recent eDNA agriculture application experiments, the fragmentation of eDNA has been achieved with ultrasound. It can be applied directly to cellular samples or purified genomic material with very similar results. Salt concentration, exposure time, power, and temperature can be manipulated to control the length and potentially the form of fragment desired (single- or double-stranded) [107].

## 6. Perspectives

As we have shown before, nucleic acids have strong biological potential if applied as treatment in agricultural systems (as DAMP or MAMP/PAMP), depending on the concentration and origin (self or non-self), generating diverse reactions on plants [46,47,52,68].

As reported before [47,60], the plant immune system responds to the detection of both self and non-self eDNA. Due to the effect on plant metabolism seems to be different between eDNA sources, these results suggest a role of eDNA different from the suspected DAMP or MAMP (PAMP) role as a biological plant biostimulant (considering a hormetic effect), activating several metabolic pathways related to both defence and growth processes depending on the dose and the opportunity of applications during plant phenology.

Here, we suggest that if eDNA from different sources is applied to plants in the right concentration, it might induce several biological pathways related on one hand to defence mechanisms (elicitation effect), and on the other hand with growth and development (biostimulant effect), a feature of some elicitors based on hormetic behaviour [108]. Precisely, all these aspects should be studied in more detail to design adequate applications strategies for eDNA in agriculture.

It would be highly interesting to experimentally evaluate the extraction and application of eDNA from different sources and mixtures as a potential or elicitor based on a hormetic study. As it has been said, vegetable waste in the horticultural industry can be an important source of macronutrients [109], and its poor management can lead to some environmental issues. The eDNA can be extracted from agricultural solid wastes making agricultural production more sustainable and closer to a circular economy system.

Agriculture production is continuously increasing and generates great amounts of waste annually; in fact, this has been a challenging problem worldwide [110]. Worldwide average production in 2018–2020 rounded 2413.4 Mtons only for maize, wheat, and rice [111]. From this industry, waste has been estimated as 20% of the entire production [112]. Some options for residue management are currently applied around the world, it has been reported that vegetal residues are commonly used for animal feeding, fungi cultivation, composting process, industrial uses, bioenergetics, etc. [113]. It has been estimated that around 30–60% of organic waste is collected and reused in the European Union, but other world areas are still wasting this material, as the United States and Canada reuse 32% and Latin American countries only 7% [114].

Nowadays, worldwide, the level of organic waste treatment and reuse represents a great opportunity to develop productive technologies from this material. DNA can be extracted from this kind of waste, especially from the agricultural waste of each production. Additionally, vegetal waste material would present some level of decay and possibly the nucleic acids inside are already fragmented in some way, possibly in fragment sizes enough to satisfy the agricultural purposes as biostimulant/elicitor, lowering the management cost even more. As shown before, the concentration of applied eDNA in crops represents one of the most challenging variables to standardize in an agriculture treatment. This increases the importance to determine the minimum hormetic concentration needed depending on the application goal. Once several evaluations have covered a wider dose-effect spectrum, researchers will be able to design different agricultural applications of eDNA concentrations, assuring to stay in the eustress zone of hormetic curve obtaining a biostimulant effect, as explained by Vargas-Hernandez et al. [27]. This potential application would also increase the importance of a hazardous biological material-free and toxic chemical-free DNA extraction; the best strategy would be extraction from organic waste that can be then coupled with a composting process. The design of a controlled elicitation strategy in crop production using eDNA must include the study of variables as concentration (for self eDNA normally the literature reports low concentrations as 2 ppm), methodology of application (i.e. drench or spraying), plant stage, number of applications during cultivation, etc. Once we increase the knowledge of the aforementioned strategy for eDNA applications in agriculture, we could be in a clearer position to apply an industrial scale model as a perspective for the use of eDNA in agriculture depending on the plant species.

## 7. Conclusions

The evolution of agricultural practices has significantly increased crop yields by the application of improved crops, mechanical ploughing, chemical fertilizers, and pesticides [16], but recently, a negative face of traditional agricultural systems has been identified, this related to a negative environmental, ecological and health impact. Later, organic agriculture has emerged as a group of diverse green techniques with the great challenge of meeting the needs of an everyday growing world population in a matter of quantity, yield, food quality, nutritional benefit, efficient management of plant pests, and diseases and reducing the environmental impact of technological change [6,10].

Recent studies have identified the use of elicitors or plant biostimulants as a sustainable agriculture input that covers all the mentioned needs on some levels [115,116,117,118,119,120]. These inputs consist of the exposition of plant crops to certain doses of stress that had been previously evaluated by the determination of hormetic curves [121,122].

In the present review, we summarized evidence of the great potential that eDNA has as an elicitor/biostimulant by multiple natural mechanisms. This has been noted before without facing the technical challenges of practical development of this technology in agricultural systems. Here we aimed to review the viable protocols for DNA extraction and fragmentation based on the real needs of agricultural application, setting aside the traditional lab DNA extraction that immediately comes to mind. Finally, we have identified some steps of DNA extraction that can be lowered in intensity or even avoided to fulfil the agricultural needs of eDNA application as we search for fragmented DNA instead of looking for molecule integrity as commonly intended. We also suggested a more sustainable application of eDNA extraction: obtaining the nucleic acids from agricultural residues and applying self eDNA as a DAMP or mixed sources DNA as a general biostimulant/elicitor (Figure 1). This would allow producers to diminish the costs of treatment and increase the viability of the upscaling of this technology.

This review aimed to contribute, based on the biological evidence reported, in mentioning several important considerations, and take into account the use of eDNA on an industrial scale for real agricultural production systems. Although several evaluations are still needed, the application of eDNA to crops will contribute to the sustainability of agricultural systems by using natural mechanisms as treatments.

## Figures and Tables

**Figure 1 biology-10-01022-f001:**
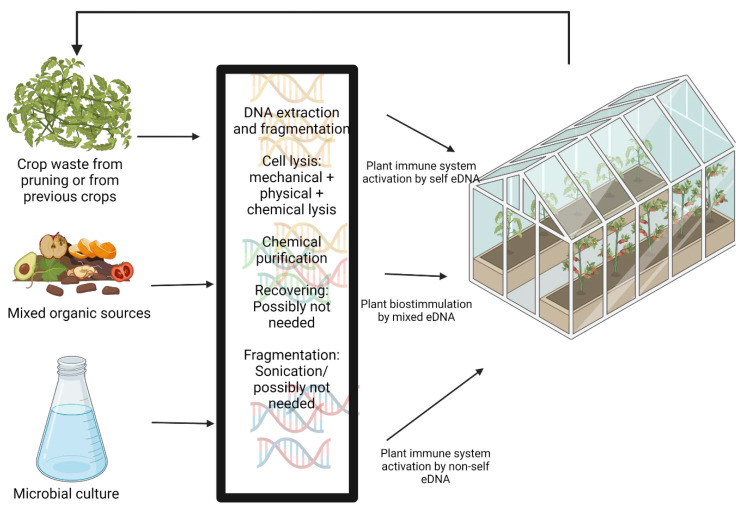
A proposed system of eDNA as an agriculture treatment with circular economy agricultural eDNA treatments. This figure was created using BioRender (https://biorender.com/, (accessed on 12 July 2021)).

**Table 1 biology-10-01022-t001:** Activation of TLRs in mammals [41,42,43].

TLR	Activation
TLR7	Respond to bacterial and viral single-stranded RNA (ssRNA), and are also activated by imidazoquinolines and other small synthetic immunomodulatory compounds
TLR8	
TLR9	Activated by DNA of viruses or bacteria with unmethylated CpG dinucleotides.

**Table 2 biology-10-01022-t002:** Summary of effects of applied eDNA experimentally evaluated.

Reference	Plant	Source of DNA	Concentration (ppm)	Effect
[67]	*Arabidopsis thaliana*	*Escherichia coli*	500	H_2_O_2_ induction and growth inhibition, callose deposition, induced expression of FRK1
[45]	*Acanthus mollis*	*Acanthus mollis*	2	No effect
			20, 200	Reduction in root growth
		*Arabidopsis thaliana, Quercus ilex, Sarcophaga carnaria*	200	No effect
[51]	*Phaseolus lunatus*	*S. littoralis oral secretions and larvae, Zea mays*	200	No effect
	*Zea mays*	*S. littoralis oral secretions and larvae, Phaseolus lunatus*	200	No effect
		*Zea mays*	2	No effect
			12, 90, 120	Increase in plasma membrane potential depolarization
			200	Increase in plasma membrane potential depolarization and Ca^2+^
	*Phaseolus lunatus*	*Phaseolus lunatus*	2, 20, 90, 120	Increase in plasma membrane potential depolarization
			200	Increase in plasma membrane potential depolarization and Ca^2+^
[52]	*Phaseolus vulgaris*	*Phaseolus vulgaris*	2, 20	No effect
			50, 100, 150, 250	Root growth inhibition
			200	Root growth inhibition, H_2_O_2_ increase, activation of MAPKs, induction of extrafloral nectar, lower infection rates by *P. syringae*.
		*Phaseolus lunatus*	200	Root growth inhibition, activation of MAPKs, lower infection rates by *P. syringae*.
		*Acacia farnesiana*	200	Lower infection rates by *P. syringae*.
[46]	*Lactuca sativa*	*Lactuca sativa*	2	Root growth inhibition, genome methylation reduction, induced expression of sod and cat
			20	Root growth inhibition, genome methylation reduction, induced expression of *sod, cat and pal*
			50, 100, 150	Root growth inhibition, genome methylation reduction, induced expression of pal
			200	Root growth inhibition, induced expression of *sod, cat*, and *pal*
		*Acaciella angustissima*	2, 20, 50, 100, 150	No effect
			200	Genome methylation reduction, induced expression of *pal*
		*Capsicum chinense*	2	No effect
			20, 50	Inhibited root growth
			100, 150	Inhibited germination and root growth
			200	Inhibited germination, genome methylation, induced expression of sod and cat
[68]	*Capsicum annum*	*P. capsici, F. oxysporum and R. solani mixed*	20, 60, 100	Resistance to pathogens and increase of total phenols and flavonoids
[53]	*Arabidopsis thaliana*	*Arabidopsis thaliana*	150	MPKs, ROS and Ca^2+^ signalling, SA and JA related genes expression upregulation, increase in H_2_O_2_ and callose accumulation, resistance against pathogens
		*Brassica oleracea*	150	Upregulation of MPK3, OXI1, and CML37 gene expression
		*C. aurantrum, Solanum lycopersicum, S. oleraceae*	150	No effect
		*Phaseolus vulgaris*	150	Upregulation of MPK3, OXI1, and CML37 gene expression
		*Zea mays*	150	Upregulation of MPK3 and OXI1 genes
[47]	*Arabidopsis thaliana*	*Arabidopsis thaliana*	200	Differential expression of less than 2.5% of total genes (upregulation: brassinosteroids and cytokinins, downregulation: abscisic acid and gibberellins)
		*Clupea harengus*	200	Differential expression of more than 15% of total genes (upregulation: salicylic acid, downregulation: abscisic acid and auxins)
[60]	*Solanum lycopersicum*	*Solanum lycopersicum*	50	Plasma transmembrane potential depolarization, ligand-gated K^+^ channels, and H_2_O_2_ production activationPlasma activation
			100	
			200	Plasma transmembrane potential depolarization, ligand-gated K^+^ channels, and H_2_O_2_ production activation, downregulation: Myo-inositol, NO, ROS, cell wall, JA and sucrose biosynthetic and metabolic process, upregulation: oxygen transport, defence responses to gram-negative bacteria, lactate and adenine biosynthetic process, auxin influx

**Table 3 biology-10-01022-t003:** Comparative characteristics of cell lysis techniques.

Technique	Advantages	Disadvantages
Mechanical	Several devices are already commercially available at an industrial scale [88]. Suitable for several kind of tissues with high efficiency [89].	Production of small cell debris so next purification steps become harder [88]. High capital investment and energy costs [89].
Physical	It has shown high efficiency [90,91].	Some methods are expensive and so it is not widely used for macroscale application [88].
Chemical	Use effective buffers that also protects DNA [90].	Must be coupled with other techniques [90]. Some chemicals are toxic and need special waste disposition [92].
Biological	High specificity. Use of enzymatical products to lyse cell wall and membrane components [90].	Must be coupled with other techniques [90]. Depending on the needed enzymes it could be expensive in bigger scales [89,93].

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
