# Peer review of "Extracellular DNA: Insight of a Signal Molecule in Crop Protection"

_biology, 2021, doi:10.3390/biology10101022_

Round 1

Reviewer 1 Report

Dear authors,

You re-submit your original manuscript to "Biology" after revision. The main topic and claim of your review is, that eDNA could be used in plant protection as an alternative to pesticides or as another bioeffector / biostimulant / elicitor to increase plant health.

Despite the fact, that eDNA has some very interesting features which have been reported so far, the authors main claim remained on the possible practical application. With such a claim the following topics needs to be addressed

  1. Extraction procedure. This must be scalable and economic. The extraction processes described are all on lab level. No estimate is given what the costs would be to produce larger quantities. The methods described are also lab level. The authors still do not described, if it is possible on indiustrial scale. I am wondering, if an extraction is needed at all. The authors should discuss, if a simple plant extraction method (as it is used widely) would not extract DNA too in a sufficient and reliable way. Why is the extraction process needed anyway, when in a simple extract DNA is contained and other beneficial compounds too. This would avoid the complex process of DNA extraction, makes application much easier (no expensive equipment needed)
  2. Which amount is needed per hectare. In their response the give a calculation for tomato and that the eDNA should be produced from the same source. This has several implications like how much material is available and when, the logistics behind, how many applications are envisaged per cultivation to mention only a few. It could be an interesting case study though, when the authors would collect data on teh residues in tomato production, when these residues orccur, what happend to these residues now and if the use for eDNA could be superior then.
  3. Which effect can be achieved at which concentration of DNA. Especially which effects on plant health could be achieved. A summarizing table on this would be much better than a long description where information is somewhat "hidden".

In your introduction (L 45-84) you describe the accepted knowledge on pouplation growth, climate change, agriculture, and how breeding have changed plant traits over the past (L87-95). While I agree to all of your arguments on these topics, I do not see how this contributes to your topic on eDNA. It could be a more simple way: every new and more sustainable approach in agriculture would be welcomed.

The chapter on "Natural conditions of eDNA release and sensing" is on natural occurence of eDNA. I miss somewhat the relation to the context of application of eDNA in contrast to natural occurence. Is this a contrasting effect? The sensing is dealing first with mammals, but the review on plants is short.

The next chapters are covering eDNA as DAMP and MAMP before in chapter 5 "Technical challenges of eDNA application as an agricultural treatment" are described. I still found chapter 5 weak. I miss an in depth discussion about the extraction, why it is needed. What makes the extraction of DNA so important.

Chapter 5 is critical for me. I have expressed my major concerns on it here and in the attached PDF. The authors in their response to the first peer-review of their manuscript could still not convince me that the approach of DNA extraction will be successfull or is an interesting new way to get to DNA. (I will send my response to the editior, because I was unable to upload two files)

It should be clarified, if DNA must be extracted or that plant extracts (containing DNA) are not as good as purified DNA. I did not find any discussion on this topic

In chapter 6 "PERSPECTIVES" the authors tried to span from effects of eDNA on plants to possible application in agriculture. But the discussion again stops at the point of "DNA extraction" and does discuss sufficiently possoble applications.

In summary, The manuscript falls apart into two section:

  1. Effects of eDNA, which delivers interesting information and insights, but still in many cases no in depth discussion about the meaning for plant health.
  2. The idea of agricutural application. This section remains speculative for me. This is due to the idea of the authors, that DNA extraction and further processing is required. I still doubt that the suggested approach is successful. The authors do not present evidence that such an approach would be a way in agriculture. Or that it is superior in comparison to other plant extracts.

A more general issue is not addressed. The importance of the source for the DNA. Can it be any biological source, must it be defined, what happend with mixtures...

The review claims "agricultural applications", but this is not convincingly described. It is not giving a new insight into the topic of eDNA in agricultural practice. After reading the article the reader is left with two major outcomes: (1) eDNA have some interesting effects and (2) it is still unclear how this could be used in agriculture.

The authors have tried to summarize their ideas in figure 3 though. Taking a look at this picture a much clearer approach of the artcile would be to describe

  1. The effect eDNA could have (which is covered in the artcile)
  2. Which sources are suited
  3. Which process is required and I would add
  4. critical evaluation of economics and applicability

From these topics only (1) has been addressed sufficiently (although not as clear as in figure 3). I have read now several times the manuscript and in the end I come to a very simple question: "Why not using a simple plant extract?" or to put it differently "What is the major advantage of having an extracted and purified eDNA?"

I may have redundantly addressed the same topics and reapeted my concerns. I still have tried to address my major concerns. Despite the fact that I find it an interesting idea to use eDNA, the manuscript is not suited for publication still.

Author Response

Addressing Reviewer 1 comments:

Dear reviewer,

We are grateful for your comments, which have been addressed and attended in the best way we could. We agree with your concerns about the whole idea of the agricultural application of eDNA. In this document we respond to each comment and hope to clarify a bit your concerns. The changes are indicated in each response and in the document

Sincerely,

Dr. Ramon Gerardo Guevara González, on behalf of my coauthors

  1. A Biosystems Engineering, Autonomous University of Queretaro, Santiago de Queretaro, Mexico.

[email protected] (R.G.G-G)

Manuscript ID

biology-1385858

Type

Review

Title

Extracellular DNA: insight of a signal molecule in agricultural crop protection applications.

Author's Reply to the Review Report (Reviewer 1)

You re-submit your original manuscript to "Biology" after revision. The main topic and claim of your review is, that eDNA could be used in plant protection as an alternative to pesticides or as another bioeffector / biostimulant / elicitor to increase plant health.

Despite the fact, that eDNA has some very interesting features which have been reported so far, the authors main claim remained on the possible practical application. With such a claim the following topics needs to be addressed

Extraction procedure. This must be scalable and economic. The extraction processes described are all on lab level. No estimate is given what the costs would be to produce larger quantities. The methods described are also lab level. The authors still do not described, if it is possible on industrial scale.

  1. We have added two paragraphs regarding industrial scale of DNA extraction suggesting that it is possible to extract DNA in the amount needed for agricultural application.

I am wondering, if an extraction is needed at all. The authors should discuss, if a simple plant extraction method (as it is used widely) would not extract DNA too in a sufficient and reliable way. Why is the extraction process needed anyway, when in a simple extract DNA is contained and other beneficial compounds too. This would avoid the complex process of DNA extraction, makes application much easier (no expensive equipment needed).

  1. This is an issue that we have addressed from the start. As kindly the reviewer mention in this comment: “a simple plant extraction method (as it is used widely) “, this comment sounds similar as the way in which Plant Derived Biostimulants (PDBs) are currently and widely produced. Although plant extracts or Plant Derived Biostimulants (PDB) are widely used as an ecological approach with functional results there are still some features that eDNA offers contrasting PDB application.
  2. First the suggested mechanism in which eDNA and PDB function are different since PDB contains several components (including eDNA) that effect the desired function on plant metabolism. On one hand, the content of PDB is difficult to standardize and thus the effect on plant, because the production of biostimulant molecules depends on plant age, used solvent for the extraction, genetic background of plant variety and the environmental conditions that rebound on second metabolite production (as mentioned in the review published by Zulfiqar et al. 2019 in Plant science, https://doi.org/10.1016/j.plantsci.2019.110194). On the other hand, DNA will always be present not depending on “environmental factors on plants” (l.440-444).
  3. About the production of plant extracts, the process is not very different or cheaper from the DNA extraction suggested on lines 529-530. For the suggested methodology, the major concerns of industrial scale up is the tissue lysis which is basically one of the obligated processes on plant extracts production, and centrifugation which is already industrially up scaled. For plant extracts (as they focus on specifically secondary metabolites) other processes are needed as drying and extraction with solvents performed in expensive equipment, and heating equipment (high energy needed) as rotary evaporators, desiccators, etc. Moreover sometimes this is also a procedure during days.
  4. As discussed on the lines 448-450 “We address the most important goals to achieve in DNA extraction as the elimination of most cellular debris in the sample that may contain lytic enzymes, biological contamination, and biological hazard for agricultural application of eDNA.” The stability of eDNA on a plant extract to allow the commercial logistics would be achieved only by freezing the sample, this would make the treatment more expensive. With a simple DNA extraction, we eliminate lytic enzymes allowing the stability of DNA at environmental temperature.

Which amount is needed per hectare. In their response the give a calculation for tomato and that the eDNA should be produced from the same source. This has several implications like how much material is available and when, the logistics behind, how many applications are envisaged per cultivation to mention only a few.

It could be an interesting case study though, when the authors would collect data on teh residues in tomato production, when these residues orccur, what happend to these residues now and if the use for eDNA could be superior then.

  1. In lines 553-562 we present an estimation of the amount needed per hectare and the available material keeping the example of tomato production, but it is clear that this is highly variable by many factors also mentioned in these lines. About what happen to the agriculture residues is difficult to know due to the several differences in agricultural systems worldwide but in lines 607-622 we resume some of these data. Agricultural systems don’t dispose their waste properly. Moreover, to envisage the number of applications is still unknown for use in agriculture, because this aspect should be studied in more detail before commercial use. For instance, in chili pepper our group reported 2 applications during the whole cycle of chili pepper grown in pots under greenhouse conditions to cope wilt rot root disease (Serrano-Jamaica et al. 2021. Frontiers in Plant Science). This is an aspect that in the near future should be studied to support the number of applications and thus the quantity of plant material we need to process to satisfy the eDNA necessity.

Which effect can be achieved at which concentration of DNA. Especially which effects on plant health could be achieved. A summarizing table on this would be much better than a long description where information is somewhat "hidden".

  1. We added the summarizing table, we thought it is a valuable suggestion.

In your introduction (L 45-84) you describe the accepted knowledge on population growth, climate change, agriculture, and how breeding have changed plant traits over the past (L87-95). While I agree to all of your arguments on these topics, I do not see how this contributes to your topic on eDNA. It could be a more simple way: every new and more sustainable approach in agriculture would be welcomed.

  1. On this section we wanted to highlight the potential of elicitors as eDNA to act not only as immune system elicitors but as biostimulants too, and the importance of performing hormetic behavior characterization on each case to achieve this goal. Also we eliminate the lines 47-53 and 81-86 of the initial manuscript, which we agree with you that were repetitive and unnecessary.

The chapter on "Natural conditions of eDNA release and sensing" is on natural occurrence of eDNA. I miss somewhat the relation to the context of application of eDNA in contrast to natural occurence. Is this a contrasting effect? The sensing is dealing first with mammals, but the review on plants is short.

  1. The natural occurrence of eDNA define some features of the molecule sensed by the plant such as concentration, fragmentation and source and thus define the mechanisms activated in plant once the molecule is sensed. We think this is important to understand the expected effects on plant metabolisms when eDNA is applied and to choose the features in applied eDNA.

The next chapters are covering eDNA as DAMP and MAMP before in chapter 5 "Technical challenges of eDNA application as an agricultural treatment" are described. I still found chapter 5 weak. I miss an in depth discussion about the extraction, why it is needed. What makes the extraction of DNA so important.

Chapter 5 is critical for me. I have expressed my major concerns on it here and in the attached PDF. The authors in their response to the first peer-review of their manuscript could still not convince me that the approach of DNA extraction will be successfull or is an interesting new way to get to DNA. (I will send my response to the editior, because I was unable to upload two files)

  1. We responded above in previous similar comment

It should be clarified, if DNA must be extracted or that plant extracts (containing DNA) are not as good as purified DNA. I did not find any discussion on this topic

  1. We responded above in previous similar comment

In chapter 6 "PERSPECTIVES" the authors tried to span from effects of eDNA on plants to possible application in agriculture. But the discussion again stops at the point of "DNA extraction" and does discuss sufficiently possoble applications.

  1. We think this comment is similar to the ones previously we addressed.

In summary, The manuscript falls apart into two section:

Effects of eDNA, which delivers interesting information and insights, but still in many cases no in depth discussion about the meaning for plant health.

The idea of agricutural application. This section remains speculative for me. This is due to the idea of the authors, that DNA extraction and further processing is required. I still doubt that the suggested approach is successful. The authors do not present evidence that such an approach would be a way in agriculture. Or that it is superior in comparison to other plant extracts.

A more general issue is not addressed. The importance of the source for the DNA. Can it be any biological source, must it be defined, what happend with mixtures...

  1. We think that the description of every way in which DNA can function as an elicitor clarify the sources of extraction (DAMP: same plant, PAMP: microbial cultures), we have suggested the agricultural residues such as pruning waste would function as source. Regarding mixtures we addressed that option as a perspective that needs to be evaluated experimentally (l.596-597), to date only one experimental evidence of DNA mixtures effect on plants is available and the mixture consist of DNA extracted from 3 pathogenic microbes and the effect was successfully demonstrated (Serrano-Jamaica et al. 2021 DOI: 10.3389/fpls.2020.581891).

The review claims "agricultural applications", but this is not convincingly described. It is not giving a new insight into the topic of eDNA in agricultural practice. After reading the article the reader is left with two major outcomes: (1) eDNA have some interesting effects and (2) it is still unclear how this could be used in agriculture.

The authors have tried to summarize their ideas in figure 3 though. Taking a look at this picture a much clearer approach of the artcile would be to describe

The effect eDNA could have (which is covered in the artcile)

Which sources are suited

Which process is required and I would add

critical evaluation of economics and applicability

From these topics only (1) has been addressed sufficiently (although not as clear as in figure 3). I have read now several times the manuscript and in the end I come to a very simple question: "Why not using a simple plant extract?" or to put it differently "What is the major advantage of having an extracted and purified eDNA?"

  1. We addressed this comment above and described in the manuscript.

I may have redundantly addressed the same topics and repeated my concerns. I still have tried to address my major concerns. Despite the fact that I find it an interesting idea to use eDNA, the manuscript is not suited for publication still.

  1. Thanks for all the comments. We hope this new version has been improved as you suggest.

Respectfully,

Dr. Ramon G. Guevara-Gonzalez

Reviewer 2 Report

The manuscript has improved significantly from the first version which I have reviewed before for another MDPI journal. Very importantly, also the issues concerning non-toxicity and biological safety have been properly addressed by the authors in this new version, as have been all other needed corrections.

In the new and structurally changed version of this manuscript, which I have reviewed here now, some corrections and clarifying changes have to be made. After this is done, this review article is fit for publication in ´Biology`, in my opinion. It provides interesting and important ideas concerning the agricultural use of eDNA within a circular economy and makes suggestions, how to further the development of this into a viable, sustainable large-scale agricultural application.

I list the needed corrections and clarification in two category-paragraphs below, firstly ´ Remarks and corrections concerning the scientific content´ and secondly “Language and writing”.

Remarks and corrections concerning the scientific content:

l. 258-259: Importantly, PR1, PDF1.2, VSP2, ERF2 and ERF5 are NOT ”hormone biosynthesis genes”. The are used as marker genes for the activation of specific hormone-activated signaling pathways, PR1 for salicylic acid-activated signaling and PDF1.2 as well as VSP2 for both, jasmonic acid-activated signaling and ethylene activated signaling. Please correct this. (Of all genes listed by you only JAR1 is a biosynthesis gene, involved in JA-Ile biosynthesis.) Please write all gene names, also when you talk about ”expression of genes” in italics.

l. 309: “TIR-NB-LRRs”, since this is a whole group of R-genes.

l. 321: “remarkable differential gene expression in non-self DNA”: The word “differential” makes the scientific content here unclear: Differential between what situations/treatments? Please clarify and rephrase.

l. 326: What do you mean by “activation of the chloroplast genome”? Please clarify and specify this.

l. 352-353: As I have written already in my last review report (for `Plants´) of the older form of this manuscript, in which the same sentence in the same paragraph occurred: Please provide here references for these “previous studies” also at the end of this sentence. 

l. 394: Other PAMPS are still present. To take this into account, you could write e.g. “…more “invisible” way…” or something similar.

l. 402-403: “that can be related to the control…complex tissues rot.” This is unclear, please clarify and rephrase.

l. 527 + l. 529: Reference [100] refers to a paper about a CTAB-method; it is not an example for non-toxicity, as you yourselves write before. Please omit this reference here in this paragraph supposedly only naming mon-toxic methods (see l. 522). For l. 527-529 in its current phrasing you would have to find then a new, suitable reference or omit this sentence completely.

l. 546: “some studies”: Please provide references for these studies.

Paragraph l. 552-559, dealing with results from reference [49]: Already the very long paragraph l. 300-328 deals exclusively with results from reference [49]. Please omit this second paragraph here in l. 552-559, also, because it is partially a repetition of the former paragraph.

Paragraph l. 560-572, dealing with results from reference [62]: Also here, the long paragraph l. 285-299 deals exclusively with results from this very reference [62]. Please unite these two paragraphs in such a way, that only the paragraph starting in l. 285 describes the results from [62] and you do not have later a second long paragraph about the very same reference.

l.581-582: This sentence is not understandable. Please omit it or rephrase it.

l. 597: Please specify, what you mean with “organic residues” here, since also the animal manure discussed before is an organic residue. The same applies for “organic waste” in l. 615.

Language and writing

In the following, some language and writing corrections and clarifications, given in quotation marks. A remark: Please take care not the let a second sentence start within the previous sentence but make a full stop before and start a new sentence.

l. 16: ”weeds” instead of ”herbs”

l. 20: ”such as defense...”

Also, few minor language errors have to be addressed in the abstract; they are easy to spot.

l. 89: ”this mankind process”. Please rephrase, e.g., to ”artificial process” or an expression with the word ”manmade”.

l. 90-91: ”and unintended consequences ... may have ensued, such as”

l. 142-143: ”in specific situations, relevant to their...”

l. 146: ”have addressed”

l. 221: ”with plants showing no response to the latter”

l. 250: ”plant hormones”

l. 273: ”leads to a broad range-resistance”

l. 286: “more global way”

l. 287: ”The firsts evaluated by measurement of...obtaining…” This is not understandable concerning its meaning; please rephrase.

l. 303: “transcriptome analysis was performed”

l. 314: Please write out “hpt” here in the main text body since you do not use it as a unit following a value.

l. 314-315: “of self-DNA, while…” since a main clause (= main sentence) cannot start with “While…”.

l. 316: “was revealed.”

l. 320: Please cite at the end of this sentence again “[49]” so that it becomes clear to the reader that the whole relatively long paragraph (l. 300-320) states results and discussion-points from [49].

l. 333: “pathogen attack”

l. 334: “…danger. On the other hand,…”

l. 360: “…plants. Moreover,…”

l. 361-362: “Here, we link”

l. 366-368: may be explained…but evidence suggests another explanation.

l. 373: “in light of the latter result, we suggest that DNAse I…”

l. 375 + l. 393: Please write “plant immune system” instead of “vegetal immune system”.

l. 379: “…treatment. By contrast, …”

l. 391: “dissert” means something completely different. Please rephrase.

l. 437: Should it not read “Duran-Flores”, as in the reference list? Please unify the name writing in citation and reference list here.

l. 406-407: “by the pathogenic fungus Z. tritici”

l. 420: “…affects SA signaling. As the initial…”

l. 451: “waste disposal”

l. 454: “it is important”

l.457: “with a pestle”

l. 466: “remaining”

l. 471: Please omit “As we know, …”

l. 475: “increase the cost”

l. 477: “…genetic material. In this case, extracted DNA…”

l. 482: “general tissue lysis method”

l. 493: Please change to: “techniques, mainly because” In this case it is opposite to the usual error in several other occasions. Here you seemingly start a main clause/sentence (Mainly …) where it is in fact none, it has no verb.

l. 521: “Some simple, safe handling methods with low equipment dependence and with…”

l. 524: “and applying”

l. 547: “on plants”
l.584-585: “during production and therefore achieving economic development”

l. 590: “70% of food”

l. 593: “15.862 Mtonnes /cwe” for what?

l. 605: “fragment sizes”

l. 614: “hazardous biological material-free and toxic chemical-free DNA extraction”

l. 663: “the application of eDNA to crops will contribute to the sustainability”

Author Response

Addressing Reviewer´s 1 comments:

Comments and Suggestions for Authors

Remarks and corrections concerning the scientific content:

  1. 258-259: Importantly, PR1, PDF1.2, VSP2, ERF2 and ERF5 are NOT ”hormone biosynthesis genes”. They are used as marker genes for the activation of specific hormone-activated signaling pathways, PR1 for salicylic acid-activated signaling and PDF1.2 as well as VSP2 for both, jasmonic acid-activated signaling and ethylene activated signaling. Please correct this. (Of all genes listed by you only JAR1 is a biosynthesis gene, involved in JA-Ile biosynthesis.) Please write all gene names, also when you talk about “expression of genes” in italics.
  2. Thanks for the comment and agree. We have changed this paragraph by writing the gene names in italics and changing the expression “hormone biosynthesis genes” to “marker genes regulated by defense-related phytohormones” (as described in the referenced paper).
  3. 309: “TIR-NB-LRRs”, since this is a whole group of R-genes.
  4. The correction has been made.
  5. 321: “remarkable differential gene expression in non-self DNA”: The word “differential” makes the scientific content here unclear: Differential between what situations/treatments? Please clarify and rephrase.
  6. Here we separate the results on non-self eDNA treated plants gene transcription and those from self eDNA treated plants. And clarify that both were compared to control plants with no eDNA treatment.
  7. 326: What do you mean by “activation of the chloroplast genome”? Please clarify and specify this.
  8. We meant the activation of chloroplast genes expression activation, now it is corrected in the revised manuscript.
  9. 352-353: As I have written already in my last review report (for `Plants´) of the older form of this manuscript, in which the same sentence in the same paragraph occurred: Please provide here references for these “previous studies” also at the end of this sentence.
  10. Here we have decided to erase the phrase “Together, previous studies suggest the capacity of plants to sense and identify eDNA released to the environment by microbial organisms”, as the same idea is presented later on that paragraph. Additionally, we add the references to the studies described on that section and reorder the reference number of the next paragraphs (l. 359 and 364).
  11. 394: Other PAMPS are still present. To take this into account, you could write e.g. “…more “invisible” way…” or something similar.
  12. We have changed the word “invisible” for “discreet”, we think it represents better the idea of not completely invisible about the infection (l.403)
  13. 402-403: “that can be related to the control…complex tissues rot.” This is unclear, please clarify and rephrase.
  14. The new phrase is “The microbial fragmented eDNA treatment showed an immune system activation effect that can be related to a decrease in disease severity by the inoculated pathogenic complex in capsicum tissues.” (l.411-414)
  15. 527 + l. 529: Reference [100] refers to a paper about a CTAB-method; it is not an example for non-toxicity, as you yourselves write before. Please omit this reference here in this paragraph supposedly only naming mon-toxic methods (see l. 522). For l. 527-529 in its current phrasing you would have to find then a new, suitable reference or omit this sentence completely.
  16. The references in line 538 were from protocols with simple and few number of steps and not specifically non-toxic protocols but we think that it can be confusing due to the idea that we are expressing so we removed the reference [100] from this list, but we had to keep it in line 541 because is an experimental evidence of β-mercaptoethanol successfully replaced by sodium sulphite in an extraction protocol. Researchers can take this replacement step from this work and apply it in a protocol with no CTAB use.
  17. 546: “some studies”: Please provide references for these studies.
  18. References were added in line 560, thank you for your patience on this repetitive mistake.

Paragraph l. 552-559, dealing with results from reference [49]: Already the very long paragraph l. 300-328 deals exclusively with results from reference [49]. Please omit this second paragraph here in l. 552-559, also, because it is partially a repetition of the former paragraph.

Paragraph l. 560-572, dealing with results from reference [62]: Also here, the long paragraph l. 285-299 deals exclusively with results from this very reference [62]. Please unite these two paragraphs in such a way, that only the paragraph starting in l. 285 describes the results from [62] and you do not have later a second long paragraph about the very same reference.

  1. Several changes were made to these paragraphs (l. 288-316) results from self and non self eDNA have been united by reference and resumed so the paragraph is not that long.

Also, these references were only mentioned in the non-self eDNA section (l-572-576).

l.581-582: This sentence is not understandable. Please omit it or rephrase it.

  1. Thanks and agree with your comment. We have rephrased this sentence. l. 608-610.
  2. 597: Please specify, what you mean with “organic residues” here, since also the animal manure discussed before is an organic residue. The same applies for “organic waste” in l. 615.
  3. We cleared that those are vegetal organic residues and vegetal organic waste (l.625-627).

Language and writing

In the following, some language and writing corrections and clarifications, given in quotation marks. A remark: Please take care not the let a second sentence start within the previous sentence but make a full stop before and start a new sentence.

  1. All writing issues were located and corrected, we are grateful for this detailed review is very helpful for us.
  2. 16: ”weeds” instead of ”herbs”
  3. 20: ”such as defense...”

Also, few minor language errors have to be addressed in the abstract; they are easy to spot.

  1. 89: ”this mankind process”. Please rephrase, e.g., to ”artificial process” or an expression with the word ”manmade”.
  2. 90-91: ”and unintended consequences ... may have ensued, such as”
  3. 142-143: ”in specific situations, relevant to their...”
  4. 146: ”have addressed”
  5. 221: ”with plants showing no response to the latter”
  6. 250: ”plant hormones”
  7. 273: ”leads to a broad range-resistance”
  8. 286: “more global way”
  9. 287: ”The firsts evaluated by measurement of...obtaining…” This is not understandable concerning its meaning; please rephrase.
  10. 303: “transcriptome analysis was performed”
  11. 314: Please write out “hpt” here in the main text body since you do not use it as a unit following a value.
  12. 314-315: “of self-DNA, while…” since a main clause (= main sentence) cannot start with “While…”.
  13. 316: “was revealed.”
  14. 320: Please cite at the end of this sentence again “[49]” so that it becomes clear to the reader that the whole relatively long paragraph (l. 300-320) states results and discussion-points from [49].
  15. 333: “pathogen attack”
  16. 334: “…danger. On the other hand,…”
  17. 360: “…plants. Moreover,…”
  18. 361-362: “Here, we link”
  19. 366-368: may be explained…but evidence suggests another explanation.
  20. 373: “in light of the latter result, we suggest that DNAse I…”
  21. 375 + l. 393: Please write “plant immune system” instead of “vegetal immune system”.
  22. 379: “…treatment. By contrast, …”
  23. 391: “dissert” means something completely different. Please rephrase.
  24. 437: Should it not read “Duran-Flores”, as in the reference list? Please unify the name writing in citation and reference list here.
  25. 406-407: “by the pathogenic fungus Z. tritici”
  26. 420: “…affects SA signaling. As the initial…”
  27. 451: “waste disposal”
  28. 454: “it is important”

l.457: “with a pestle”

  1. 466: “remaining”
  2. 471: Please omit “As we know, …”
  3. 475: “increase the cost”
  4. 477: “…genetic material. In this case, extracted DNA…”
  5. 482: “general tissue lysis method”
  6. 493: Please change to: “techniques, mainly because” In this case it is opposite to the usual error in several other occasions. Here you seemingly start a main clause/sentence (Mainly …) where it is in fact none, it has no verb.
  7. 521: “Some simple, safe handling methods with low equipment dependence and with…”
  8. 524: “and applying”
  9. 547: “on plants”

l.584-585: “during production and therefore achieving economic development”

  1. 590: “70% of food”
  2. 593: “15.862 Mtonnes /cwe” for what?
  3. 605: “fragment sizes”
  4. 614: “hazardous biological material-free and toxic chemical-free DNA extraction”
  5. 663: “the application of eDNA to crops will contribute to the sustainability”
  6. All writing issues were located and corrected, we are grateful for this detailed review is very helpful for us.

Respectfully,

Dr. Ramon G. Guevara-Gonzalez

Round 2

Reviewer 1 Report

Dear authors,

you have improved the manuscript in comparison to the last revsion and you have responded to all my concerns. In the newly revised manuscript I have four additional concerns which could be addressed in a minor revision.

  1. Table 3: It seems that you are a bit over optimistic in your evaluation of the different processes. The many "+" for some processes seem too much for me. Please re-evaluate your decsison for the different processes and simplify the result. I have made suggestion directly in the table.
  2. Figures 1 and 2: They are still not more that placeholders. They do not provide any aditional information. Both are very simplistic and do not reflect the "real" complexity. I strongly suggest to delete both!
  3. You newly added text in L532-558 is a good addition, but I have made some comments. In some points you should be more specific. Because you are focussing here on tomato (which I think is a good example), you need to make a reference to the works of Heuvelink and De Koning (see my comments on this).
  4. I suggest a slightly different title.

You have done now two revisions and the improvements made are well recognized. With these final small revisions the article would be ready for publication and open to the scientific discussion.

Author Response

Amazcala, El Marques, Queretaro, México; 27-September-2021

Dear Editor

Biology

Enclosed this cover letter I am sending the minor revised manuscript (Manuscript ID-biology -1385858), entitled, “Extracellular DNA: insight of a signal molecule in agricultural crop protection applications”, whose authors are: Ireri Alejandra Carbajal-Valenzuela , Gabriela Medina-Ramos , Laura Helena Caicedo-López , Alejandra Jimenez-Hernandez , Adrian Esteban Ortega-Torres , Luis Miguel Contreras-Medina , Irineo Torres-Pacheco and Ramon Gerardo Guevara-Gonzalez. The corrections suggested by reviewer 1 were addressed and indicated below and in the revised version of the manuscript:

Author's Reply to the Review Report (Reviewer 1)

Please provide a point-by-point response to the reviewer’s comments and either enter it in the box below or upload it as a Word/PDF file. Please write down "Please see the attachment." in the box if you only upload an attachment. An example can be found here.

* Author's Notes to Reviewer

Dear authors,

You have improved the manuscript in comparison to the last revision and you have responded to all my concerns. In the newly revised manuscript I have four additional concerns, which could be addressed in a minor revision.

Table 3: It seems that you are a bit over optimistic in your evaluation of the different processes. The many "+" for some processes seem too much for me. Please re-evaluate your decision for the different processes and simplify the result. I have made suggestion directly in the table.

Response: The table has been changed, now only the evaluated characteristics are mentioned per technique and references have been added. Thank you for your kind comment.

Figures 1 and 2: They are still not more that placeholders. They do not provide any additional information. Both are very simplistic and do not reflect the "real" complexity. I strongly suggest to delete both!

Response: The images have been deleted

You newly added text in L532-558 is a good addition, but I have made some comments. In some points you should be more specific. Because you are focusing here on tomato (which I think is a good example), you need to make a reference to the works of Heuvelink and De Koning (see my comments on this).

Response: We found your comments on this issue very helpful, we have added lines 594-599 where we cite the references that you kindly shared in your comments.

I suggest a slightly different title.

Response: The tittle has been changed as suggested.

You have done now two revisions and the improvements made are well recognized. With these final small revisions the article would be ready for publication and open to the scientific discussion.

We are very thankful for your comments and perspectives about our review, we hope this new minor revised version is OK for publication.

Respectfully,

Dr. Ramón Gerardo Guevara González
